# Flat-Top Line-Shaped Beam Shaping and System Design

**DOI:** 10.3390/s22114199

**Published:** 2022-05-31

**Authors:** Che Liu, Yanling Guo

**Affiliations:** College of Mechanical and Electrical Engineering, Northeast Forestry University, Harbin 150040, China; liuche1988@gmail.com

**Keywords:** selective laser sintering, flat-top line beam, dynamic focus

## Abstract

In this study, the circular Gaussian spot emitted by a laser light source is shaped into a rectangular flat-top beam to improve the scanning efficiency of a selective laser sintering scanning system. A CO_2_ laser with a power of 200 W, wavelength of 10.6 μm, and spot diameter of 9 mm is shaped into a flat-top spot with a length and width of 0.5 × 0.1 mm, and the mapping function and flat-top Lorentzian function are calculated. We utilize ZEMAX to optimize the aspherical cylindrical lens of the shaping system and the cylindrical lens of the focusing system. We then calculate the energy uniformity of the flat-top line-shaped beam at distances from 500 to 535 mm and study the zoom displacement of the focusing lens system. The results indicated that the energy uniformity of the flat-top beam was greater than 80% at the distances considered, and the focusing system must precisely control the displacement of the cylindrical lens in the Y-direction to achieve precise zooming.

## 1. Introduction

In a selective laser sintering 3D-printing system, the emitted laser beam has a Gaussian energy distribution and circular spot. It also has a point-shaped Gaussian energy distribution after focusing; therefore, direct applications typically result in uneven heating and low-sintering-molding efficiency. To mitigate this limitation in practical applications, a laser beam with a circular spot and Gaussian energy distribution must be shaped into a rectangular spot with a flat-top energy distribution, and it should have a linear flat-top energy distribution after focusing [1,2,3]. This type of line-shaped laser irradiation sintering (line-shaped sintering) is equivalent to multiple lasers working simultaneously, and the resultant heating is uniform. This method can improve sintering quality and shorten the sintering time of molded parts. Moreover, in the process of laser shaping, the laser divergence angle is compressed to reduce the diffraction of the laser beam and obtain a thinner focusing line-shaped spot.

Current beam-shaping methods mainly include aspherical-lens systems [4,5,6,7], diffractive optical elements [8], liquid-crystal spatial light modulators [9,10], and metasurfaces and metamaterials [11,12]. Aspherical cylindrical lenses are the most effective beam-shaping method for an intense laser beam-shaping system. This method has the advantages of a good shaping effect, low energy loss, and a simple structure. Additionally, only two aspherical cylindrical lenses are typically required to realize laser-beam expansion and shaping, and many previous studies have extensively investigated these applications. A Gaussian beam can effectively be shaped into a flat-top beam [13,14,15,16]; however, the shape of the beam spot cannot be changed.

This study proposes a beam-shaping system based on aspherical cylindrical lenses. The proposed system uses the principle of the equivalent optical length of any beam between two aspherical lenses and the law of conservation of energy of the incident and outgoing laser beams to shape a circular laser spot with a Gaussian light-intensity distribution into a quasi-rectangular spot with a uniform intensity distribution. We introduce the design principle and method used for the shaping system through an example and analyze the effectiveness of the proposed method via a practical application.

## 2. Physical Model and Mapping Function of Flat-Top Beam

The simple physical model of the flat-top beam is represented by a circle function, which has the advantage of a simple form. However, it can only describe the uniform energy distribution of a flat-top beam and is unsuitable for calculating the beam transmission characteristics. Compared to other physical flat-top-beam models, the flat-top Lorentz model is the simplest for calculation. Therefore, the flat-top Lorentz model is selected as the physical model of the flat-top beam in this study to reduce the calculation complexity [17,18,19].

The light intensity function distribution of the laser beam is shown in Equation (1):(1)$$I(r)=I_{0}\exp(\frac{-2r^{2}}{r_{0}^{2}})$$

In Equation (1), *r*_0_ is the laser beam radius (mm) and *I*_0_ is the maximum light intensity (cd) of the laser beam. The light-field intensity distribution of the laser beam is shown in Figure 1.

In Figure 1, *ω*_0_ is the waist radius of the Gaussian laser beam, defined as the radius of the laser beam when the peak light intensity drops to *I*_0_/*e*^2^.

Since only the flat-top Lorentz beam can obtain the analytical solution, the flat-top Lorentz function is used as the shaping objective. The shaping model of the flat-top Lorentz beam is shown in Figure 2.

Let the intensity of the incident light be $I_{\mathrm{in}}$, the intensity of the outgoing light be $I_{\mathrm{out}}$, the projection height of any ray on the incident plane be *r*_1_, and the corresponding projection height on the outgoing plane be *r*_2_. The beam–waist radius of the incident Gaussian beam is *ω*_0,_ and the maximum radius is *r*_0_. The outgoing flat-top beam has a radius of *R*_0_. *z*_1_(*r*) and *z*_2_(*r*) are the shape functions of two aspheric surfaces. According to the law of conservation of energy, the energies contained in *r*_1_~*r*_1_ ± Δ*r*_1_ and *r*_2_~*r*_2_ ± Δ*r*_2_ are equal. The following normalization equations can be established in the Cartesian and polar coordinate systems:(2)$$f_{1}\left(x_{1},y_{1}\right)\times I_{\mathrm{in}}\left(x_{1},y_{1}\right)dx_{1}dy_{1}=f_{2}\left(x_{2},y_{2}\right)\times I_{\mathrm{out}}\left(x_{2},y_{2}\right)dx_{2}dy_{2}=1$$
(3)$$f_{1}\left(r_{1}\right)\times 2\pi I_{\mathrm{in}}\left(r_{1}\right)r_{1}dr_{1}=f_{2}\left(r_{2}\right)\times 2\pi I_{\mathrm{out}}\left(r_{2}\right)r_{2}dr_{2}=1$$
where $f_{1}$ is the entrance pupil function and $f_{2}$ is the exit pupil function, which are shown as follows:$$f\left(r_{1}\right)=\{\begin{array}{c} 1\\ 0 \end{array}\;\begin{array}{c} \leq r_{0}\\ >r_{0} \end{array}$$
$$f\left(r_{2}\right)=\{\begin{array}{c} 1\\ 0 \end{array}\;\begin{array}{c} \leq R_{0}\\ >R_{0} \end{array}$$

The intensity distribution of the incident Gaussian beam is
(4)$$I_{G}\;\left(r\right)=\frac{2}{\pi \omega_{0}^{2}}\exp\left[-2{\left(\frac{r}{\omega_{0}}\right)}^{2}\right].$$

Considering the integrability of the outgoing flat-top beam, the flat-top Lorentzian function is used to express the intensity distribution as follows:(5)$$I_{L}\left(R\right)=\frac{1}{\pi R_{0}^{2}{\left[1+{\left(\frac{R}{R_{0}}\right)}^{q}\right]}^{1+\frac{2}{q}}}$$
where *q* is the order of the flat-top Lorentzian function.

After substituting the function expressions of the Gaussian and flat-top Lorentzian beams into Equation (3), the mapping function can be obtained as follows:(6)$$1-\exp\left[-2\left(\frac{r^{2}}{\omega_{0}^{2}}\right)\right]={\left[1+{\left(\frac{R_{0}}{R}\right)}^{q}\right]}^{-\frac{2}{q}}$$

The mapping function between *R* and *r* is
(7)$$R=h\left(r\right)=\frac{R_{0}\sqrt{1-\exp\left[-2\left(\frac{r}{\omega_{0}}\right)\right]}}{\sqrt{1-{\left\{1-\exp\left[-2{\left(\frac{r}{\omega_{0}}\right)}^{2}\right]\right\}}^{q/2}}}$$
(8)$$r=h\left(r\right)=\pm \omega_{0}\sqrt{-\frac{1}{2}\ln\left\{1-{\left[1+{\left(\frac{R}{R_{0}}\right)}^{-q}\right]}^{-\frac{2}{q}}\right\}}$$

In particular, when *q*→∞, Equation (7) can be written as
(9)$$R=R_{0}\sqrt{1-\exp\left[-2\left(\frac{r}{\omega_{0}}\right)\right]}$$

Equation (9) shows that when the flat-top Lorentzian function is used as a flat-top beam distribution function, its mapping function has an analytical solution, which can facilitate ray tracing and significantly simplify the numerical calculation process. For a Galilean-type aspheric system [20], there exists
(10)$$R=-R_{0}\sqrt{1-\exp\left[-2{\left(\frac{r}{\omega_{0}}\right)}^{2}\right]}$$

The Galileo shaping system is composed of a flat concave lens and a flat convex lens, as shown in Figure 3. The convergence point generated by the Galileo-shaped structure is a virtual focus, which can avoid the air breakdown effect, and its axial size is smaller than that of the Kepler-shaped structure. Therefore, the application of the Kepler system for beam shaping requires laser power that is not too high, and the Galileo aspheric lens group can be applied to larger power.

The magnification *β* = *f*_2_/*f*_1_, where *f*_1_ is the focal length of flat-concave lens and *f*_2_ is the focal length of flat-convex lens.

Along the cross-section of the Gaussian beam, the energy is concentrated around the spot center. To obtain a flat-top beam with uniform illumination, it is necessary to diverge the rays that pass through a small aperture and concentrate the rays that pass through a large aperture. Therefore, it is necessary to obtain the relationship between the coordinates of the rays on the entrance-pupil plane and those on the image plane, which is called the mapping function.

## 3. System Design of Laser Beam Expansion and Shaping

The optical beam expansion and shaping system based on aspherical cylindrical lenses can simultaneously adjust the intensity distribution and spot shape of the laser beam. The parameters of the incident light of the shaping object used in the system design are as follows: a CO_2_ laser is used with a power of 200 W, wavelength of 10.6 μm, and spot diameter of 9 mm. The Gaussian beam is shaped into a rectangular flat-top beam with a size of 15 × 60 mm using the aspherical cylindrical lenses. The working distance is 500 mm, and the glass material is ZnSe.

### 3.1. Design of Aspherical Cylindrical Lenses

The Y-direction is consistent with the default coordinate setting in ZEMAX, and all coordinate systems in this study are the same as the default setting in ZEMAX. First, we set the wavelength and aperture. The aperture was set to 13.5 mm, and the field of view was set to 0.

Three surfaces were inserted into the lens data editor (LDE). The second surface was set as a cylindrical surface, the glass material was set as ZnSe, and the thickness was set to 6 mm. The radius of the third surface was set to infinity. The radius of the second surface, conic, 4th, 6th, 8th, and 10th order coefficients, and the thickness of the third surface was set as optimization variables. The 2nd order system was omitted to reduce the processing complexity. The 4th, 6th, 8th, and 10th order coefficients were $a_{4}=-1.279\times 10^{5}$, $a_{6}=2.878\times 10^{7}$, $a_{8}=-2.878\times 10^{9}$, and $a_{10}=1.25\times 10^{11}$, respectively.

The aperture, field of view, and wavelength were set similarly to the Y-direction, and a macro program was used to generate the evaluation function. In the macro program, the radius of the flat-top beam was changed (from 7.5 to 30 mm), and the operand was changed accordingly (from REAY to REAX). The rays converged in the X-direction; therefore, a coordinate-break surface was added to the LDE to rotate the cylindrical lens by 90° around the *Z*-axis. The 4th, 6th, 8th, and 10th order coefficients were $a_{4}=-4.816\times 10^{5}$, $a_{6}=9.016\times 10^{7}$, $a_{8}=-8.964\times 10^{9}$, and $a_{10}=3.913\times 10^{11}$, respectively.

Two cylindrical lenses were used to shape the X and Y directions, and the lenses did not interfere with each other. Thus, after the two cylindrical lenses were individually designed, they could simply be stacked. The refraction–surface radius, air thickness, nonlinear coefficient of the aspheric surface, and 4th–10th order coefficients were set as the optimization variables for the system. The distance between the shaping lenses for the X- and Y-directions was set to 142 mm. The optimized system structure is shown in Table 1.

The light-field distribution of the flat-top rectangular beam combination optical system is shown in Figure 4. Figure 4a is the light-field intensity distribution in the X direction, and Figure 4b is the light-field intensity distribution in the Y direction. From the figure, the spot size in the X direction and Y direction is 60 mm and 15 mm, respectively, which meets the design requirements. Figure 5 shows the resulting light-field distribution spot diagram of the combined optical components on the X–Y plane, where each grid division represents 5 mm.

### 3.2. Design of Focusing Lens Combination Optical Component System

#### 3.2.1. Structural Parameters of Focusing Lens Combination Optical Component System

The objective is to use aspherical cylindrical lenses to focus a rectangular flat-top beam with a size of 10 × 50 mm into a line-shaped light source with a size of 0.1 × 0.5 mm. The focus distance is 500 mm, and the glass material is ZnSe. Two aspherical shaping cylindrical lenses were designed as per the method described in Section 3.1. Based on the optimized data, the evaluation function was generated using a macro program, and the system was optimized. Table 2 shows the parameters of the optimized aspherical shaping cylindrical-lens-combined optical system. Table 3 shows the asphericity coefficient structural parameters of the optimized spherical cylindrical-lens-combined optical focusing system.

The optical structure diagrams of the focusing system in the Y- and X-directions are shown in Figure 6 and Figure 7, respectively. The geometrical dimensions of aspherical cylindrical lens 1, aspherical cylindrical lens 2, cylindrical lens 3, and cylindrical lens 4 are shown in Appendix A.

#### 3.2.2. Light-Field Distribution on the Focal Plane

An aperture diaphragm was placed 10 mm behind cylindrical lens 4. The size of the diaphragm was 20 × 4 mm, and its light-transmission efficiency was 84.472%. The flat-top distribution of the light field on the X–Y plane was more uniform after installing the diaphragm. The light-field-intensity distributions on the focal plane of the focusing system in the Y- and X-directions are shown in Figure 8 and Figure 9, respectively. The spot diagram of the light-field distribution on the X–Y plane is shown in Figure 10, where each grid division represents 0.05 mm.

#### 3.2.3. Fitting of Aspheric Coefficients of Lenses

The nonlinear coefficient of the aspheric surface given in ZEMAX had a particular error; therefore, it was necessary to use the surface sag given in ZEMAX to refit the nonlinear coefficient of the surface. The sag data were fitted using Mathematica (MathWorks).

When the 4th order coefficient was used, the fitted nonlinear coefficient was $a_{4}=5.11633\times 10^{6}$. The second aspherical lens was processed in the same way, and its aspheric coefficient was $a_{4}=5.25724\times 10^{6}$.

Table 4 shows the geometric parameters used in the calculations for each of the cylindrical lenses.

### 3.3. Study on the Zoom Function of the Lens System

It is necessary to change the focal length of the focusing lens system during the scanning and molding process of the selective laser sintering system to realize the scanning of the processing surface. In the focusing lens system, aspherical cylindrical lenses 1 and 2 shape the beams in the X- and Y-directions, respectively. Two standard cylindrical lenses, cylindrical lens 3 and cylindrical lens 4, focus the beams in the X- and Y-directions, respectively. The optical-structure diagram is shown in Figure 11. The zoom function of the system can be achieved by changing the optical interval between aspherical cylindrical lens 2 and cylindrical lens 3 and that between aspherical cylindrical lens 2 and cylindrical lens 4. Therefore, cylindrical lenses 3 and 4 are defined as a zoom lens system.

We adopted the optimized design method using the combined optical components, as described in Section 3.2, and used approximately 20% of the light-intensity difference between the center of the spot and the edge as the adjustment range. We calculated the intensity distribution of the light field and the displacement parameters of the zoom lens system with different working distances. Figure 12 and Figure 13 show some of the intensity-distribution diagrams of light fields with working distances between 500–560 mm.

When the working distance is between 500 and 535 mm, the intensity distribution of the light field exhibits a good rectangular flat-top beam. We set the working distance to a range of 500–535 mm; thus, we obtained a system scanning range of 2 L × 2 L, where
(11)$$L=\pm \sqrt{535^{2}-500^{2}}=\pm 190\;(\mathrm{mm})$$

Table 5 shows the working distance and displacement parameters of the zoom lens system.

The data in Table 5 were linearly fitted to provide a motion-control mathematical equation for the dynamic focusing. The fitted data are shown in Table 6.

The depth of field of the focusing system is ±0.7 mm. Considering the systematic and random errors caused by the subsequent mechanical and electronic systems, we set the focusing error (working distance error) of the zoom lens system to ±0.1 mm, which is 1/7 of the total error.

The fitted equation is:y_1 fitted_ = 17.61 − 0.014x(12)

y_2 fitted_ = 0.00123x^2^ − 1.64067x + 544.3596(13)

Taking the derivatives of Equations (12) and (13) and including an error of 0.1 mm in the equations, we obtain the following:
Δy_1_
_fitted_ = (−0.014) × Δx = (−0.014) × (±0.1) = ±0.0014 (mm)(14)
Δy_2 fitted_ = 0.00123Δx − 1.64064= 0.00123 × (±0.1) − 1.64064 ≈ −1.64 (mm)(15)

Based on the above analysis, the displacement distance of cylindrical lens 3 is 0.5 mm, and the displacement error is less than ±0.0014 mm. The displacement distance of cylindrical lens 4 is 12.91 mm, and the displacement error is less than ±1.64 mm. Therefore, the zoom lens system can achieve precise zooming if the displacement distance of cylindrical lens 3 is well controlled.

## 4. Experiment and Results

We performed a laser uniformity test on the designed optical system and used a CMOS beam analyzer (CinCam, CINOGY Technologies, Duderstadt, Germany) for testing. The diameter of the collimated Gaussian laser spot was approximately 9.01 mm, the size of the shaped rectangular flat-top spot was approximately 0.1 × 0.5 mm, and the laser energy in the spot was uniformly distributed.

The laser shaping and focusing system was tested using a selective laser sintering rapid prototyping machine (ASF 360, Longyuan AFS Co., Ltd., Beijing, China). Polystyrene powder produced by Longyuan AFS Co., Ltd. was used as the printing material. The flat-top line-shaped laser beam shaping system has an energy loss of approximately 17%; therefore, this study selected a 15 W Gaussian laser spot with a diameter of 0.1 mm and a 93 W flat-top line-shaped laser spot with a length and width of 0.5 mm and 0.1 mm, respectively, for experiments. The experimental parameters are shown in Table 7.

In the selective laser sintering system, the molded parts are typically placed in one of three ways: horizontally, vertically, and sideways, as shown in Figure 14. Owing to the large volume of the parts, multiple layers and a long workbench scanning time are required to print the molded parts in the vertical direction. Therefore, this study only investigated the influence of the sideways placement and horizontal placement methods on the molding speed of the molded parts. Diagrams of the experimental samples are shown in Figure 15 and Figure 16, and each independent part is a 20 × 10 × 100 mm cuboid.

The laser scanning path adopted the alternate scanning modes of the X- and Y-axes, and the experimental molded parts were placed at 0°, 30°, and 45° from the longest side in the X-axis direction, as shown in Figure 17.

Sample model 1 was placed at 0°, 30°, and 45° from the longest side in the X-axis direction, and the sintering experiment was performed five times. The scanning time of the galvanometer required to record the flat-top line-shaped spot and Gaussian spot are $T_{a1}$ and $T_{a2}$, respectively, and the times of the first and last scans are $T_{b1}$ and $T_{b2}$, respectively. We took the average of the results of the five experiments to obtain $\bar{T_{a1}}$, $\bar{T_{a2}}$, $\bar{T_{b1}}$, and $\bar{T_{b2}}$. The experimental results are shown in Table 8.

The experimental results show that when the flat-top line-shaped laser spot is used for scanning and the placement angle is 0°, the scanning efficiency is approximately five-times that of the Gaussian spot. When the placement angle is 45°, the improvement in the scanning efficiency is at its lowest.

The placement is also an important factor that affects molding efficiency. Different placement methods affect the height of the molded part along the Z-axis. The larger the number of layers, the longer the non-working time of the galvanometer, and the lower the molding efficiency.

The method follows that of the previous sample model. Sample model 2 was placed at 0°, 30°, and 45° from the longest side along the X-axis. The sintering experiment was performed five times, and the scanning time of the galvanometer was recorded as $T_{a}$. The scanning times of the galvanometer required to record the flat-top line-shaped spot and Gaussian spot are $T_{a1}$ and $T_{a2}$, respectively. The times of the first and last scans are $T_{b1}$ and $T_{b2}$, respectively. We took the average of the results of the five experiments to obtain $\bar{T_{a1}}$, $\bar{T_{a2}}$, $\bar{T_{b1}}$, and $\bar{T_{b2}}$. The experimental results are shown in Table 9.

By comparing Table 8 with Table 9, it can be concluded that when the flat-top line-shaped laser spot is used for scanning and the placement angle is 0°, the scanning efficiency is at its highest. When the placement angle is 45°, the scanning efficiency is at its lowest. Based on the data of sample models 2a and 2b, when the height of the Z-axis is 5 mm, the layer thickness is 0.1 mm, and the number of layers is 50, the non-working time of the galvanometer is similar. The larger the layer scanning area, the higher the scanning efficiency, and the smaller the influence of the placement angle.

We tested whether there was any deviation in the size of the experimental samples. We used a vernier caliper to measure and record the size of 27 experimental samples and calculated their size deviation in three directions.

Sample model 1 was used for multiple sintering experiments. When the Gaussian spot was used for the sintering experiment, the average deviation in the X-, Y-, and Z-directions were 0.35 mm, 0.50 mm, and −0.60 mm, respectively. When the flat-top line-shaped laser spot was used for the sintering experiment, the average deviation in the X-, Y-, and Z-directions were 0.30 mm, 0.40 mm, and −0.40 mm, respectively.

## 5. Conclusions

This study analyzed current laser-beam-shaping theory and systems and proposed a theory and system to produce non-imaging Gaussian laser beams and rectangular flat-top beam shaping. After discussing the laser beam-shaping theory of aspherical cylindrical lenses, we proposed the beam mapping function, called the flat-top Lorentzian function. Using ZEMAX, we designed the laser-beam expansion and shaping system and focusing system to mitigate the uneven beam energy when shaping a point light source into a surface light source. This study also analyzed the zoom lens system and it was observed that if the displacement of cylindrical lens 3 is precisely controlled, precise zooming can be achieved.

When the molded parts are placed horizontally, the overall molding efficiency is significantly improved. When the molded parts are placed sideways, the total scanning time of the galvanometer is similar to that achieved with horizontal placement. However, the powder bed fusion process requires more time, owing to the larger number of layers. Compared to the Gaussian laser, the scanning efficiency of the flat-top line-shaped laser is not considerably improved. When the number of layers is the same, a larger layer area results in a greater improvement in the scanning efficiency. The closer the placement angle of the molded part is to 45°, the lower the molding efficiency. However, as the layer area increases, the effect is smaller. It can be observed from the deviation rate of sample model 1 that the size deviations in the flat-top line-shaped beam and Gaussian beam are similar in the X- and Y-directions, and the size deviation in the flat-top line-shaped beam in the Z-direction is smaller than that of the Gaussian beam.

There are some limitations in this paper; the laser-beam expansion and shaping system and focusing system are designed for a flat-top line-shaped beam spot with a length and width of 0.5 mm × 0.1 mm. For other sizes of flat-top line-shaped beam spots, it is necessary to design laser-beam expansion and shaping systems and focusing systems with different parameters. In the future, the beam expansion and shaping system and focusing system of the adjustable-size flat-top line beam spot will be studied.

## Figures and Tables

**Figure 1 sensors-22-04199-f001:**
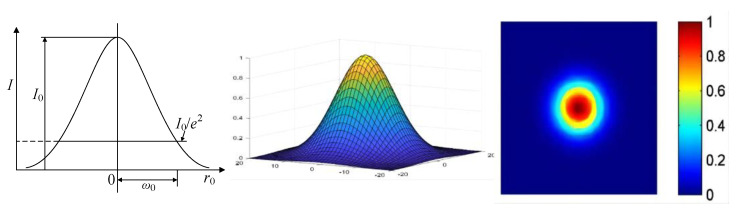
Intensity distribution of gaussian laser beams.

**Figure 2 sensors-22-04199-f002:**
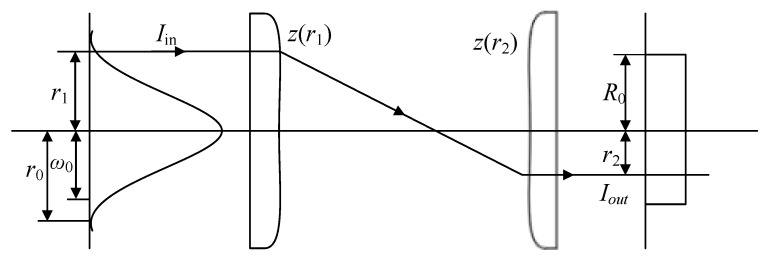
Beam-shaping model.

**Figure 3 sensors-22-04199-f003:**
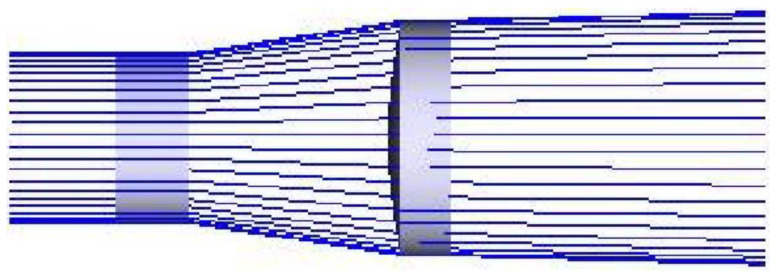
Galileo Beam-Shaping System.

**Figure 4 sensors-22-04199-f004:**
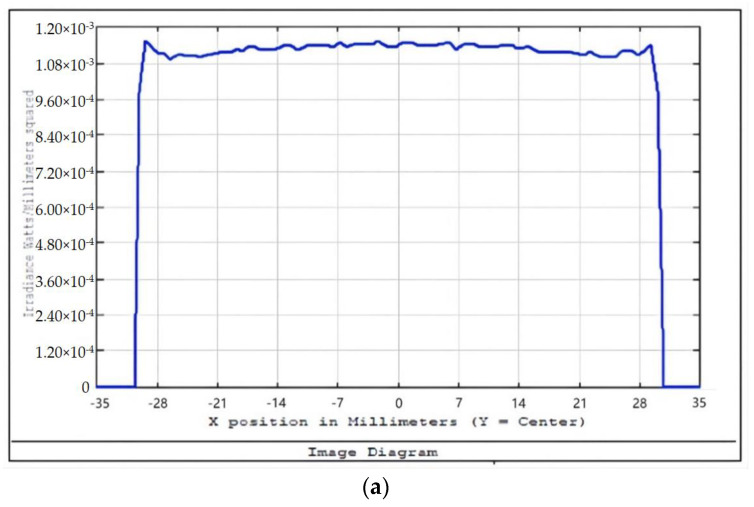
Light-field intensity distribution of the combined optical system. (**a**) The light-field intensity distribution in X. (**b**) The light-field intensity distribution in Y.

**Figure 5 sensors-22-04199-f005:**
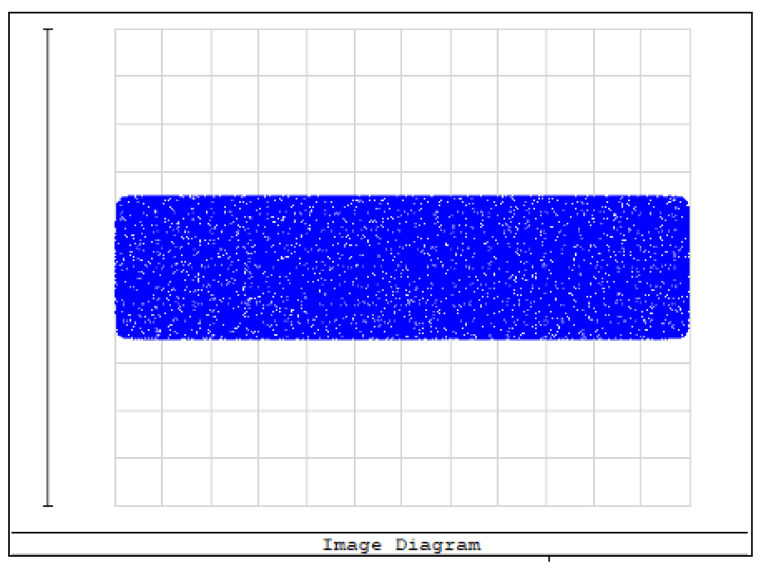
Light-field distribution spot diagram of the combined optical components on the X–Y plane.

**Figure 6 sensors-22-04199-f006:**
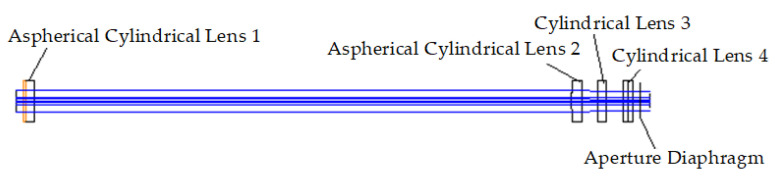
Optical structure diagram in the Y-direction.

**Figure 7 sensors-22-04199-f007:**
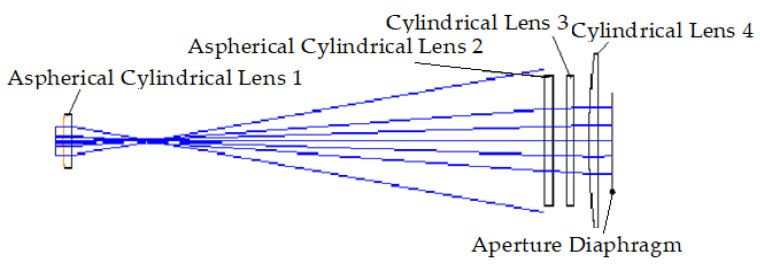
Optical structure diagram in the X-direction.

**Figure 8 sensors-22-04199-f008:**
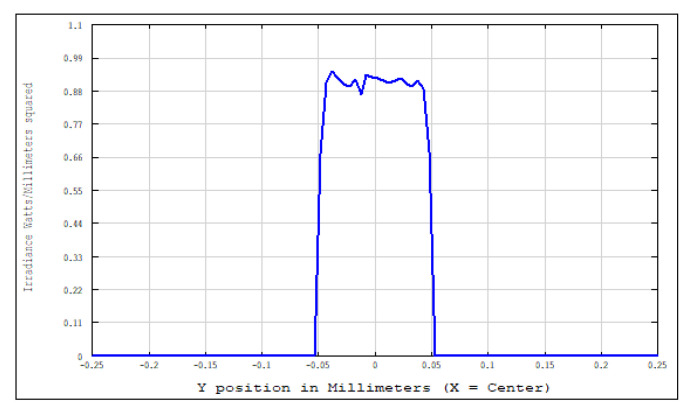
Light-field-intensity distribution in the Y-direction.

**Figure 9 sensors-22-04199-f009:**
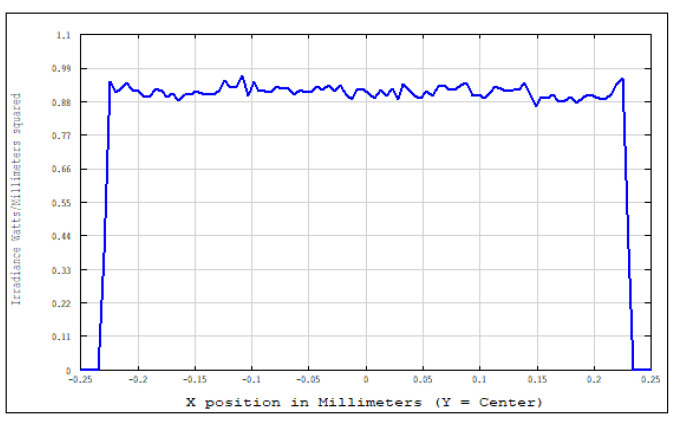
Light-field-intensity distribution in the X-direction.

**Figure 10 sensors-22-04199-f010:**
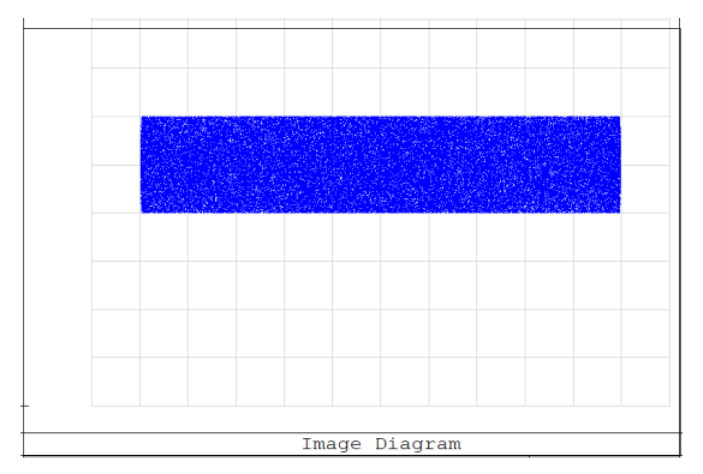
Spot diagram of light-field distribution on the X–Y plane.

**Figure 11 sensors-22-04199-f011:**
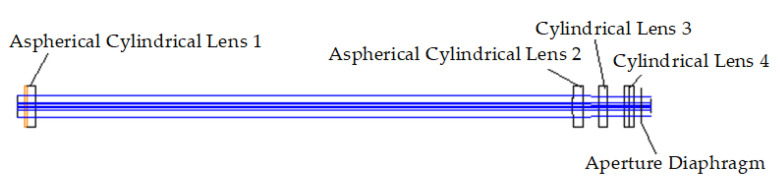
Optical structure of the focusing lens system.

**Figure 12 sensors-22-04199-f012:**
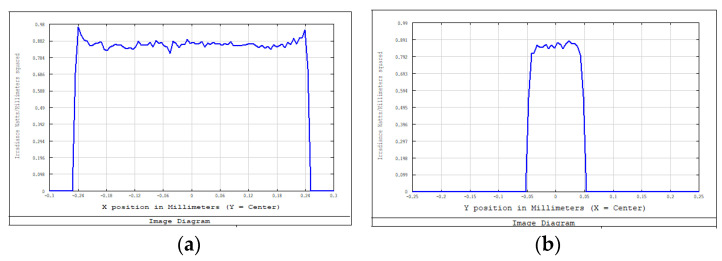
Light-field-intensity distribution of the focusing system with a working distance of 500 mm. (**a**) Light-field-intensity distribution in the X-direction. (**b**) Light-field-intensity distribution in the Y-direction.

**Figure 13 sensors-22-04199-f013:**
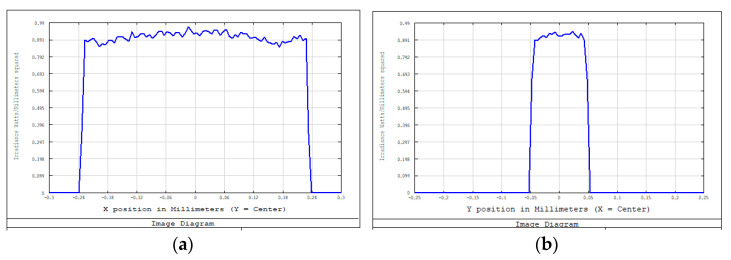
Light-field-intensity distribution of the focusing system with a working distance of 535 mm. (**a**) Light-field-intensity distribution in the X-direction. (**b**) Light-field-intensity distribution in the Y-direction.

**Figure 14 sensors-22-04199-f014:**
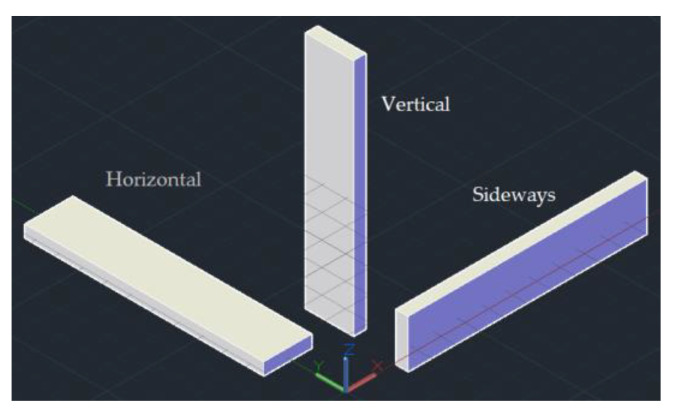
Diagram of the placement of the molded parts in three-dimensional space.

**Figure 15 sensors-22-04199-f015:**
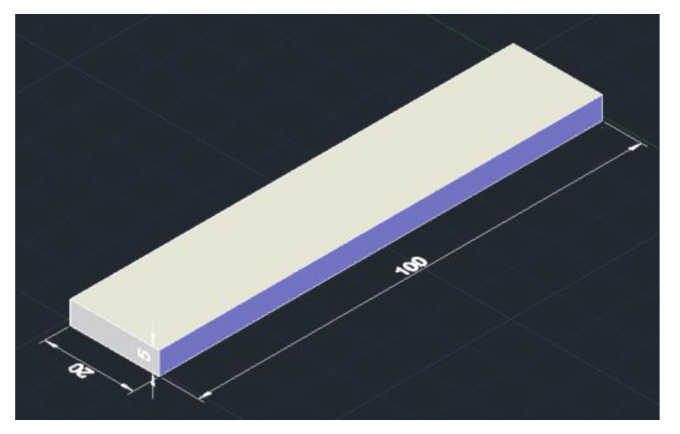
Sample model 1.

**Figure 16 sensors-22-04199-f016:**
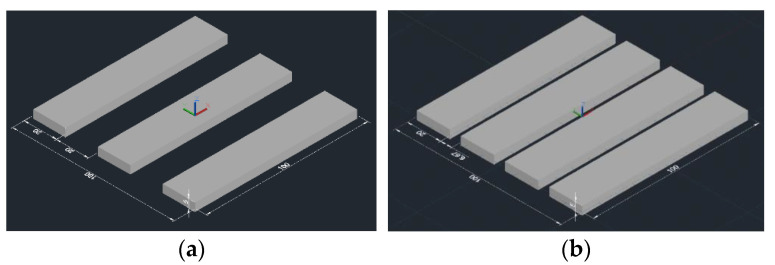
Sample model 2. (**a**) Sample model 2a. (**b**) Sample model 2b.

**Figure 17 sensors-22-04199-f017:**
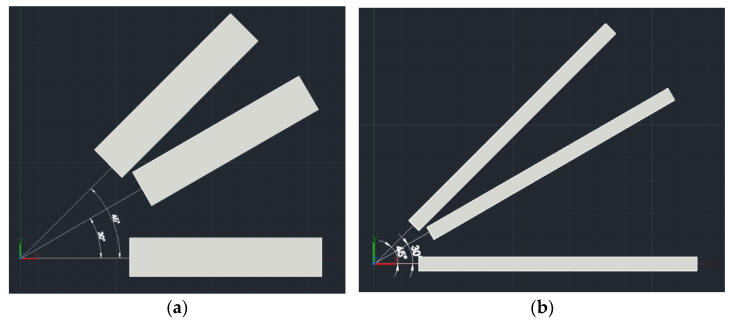
Placement of sample model 1. (**a**) Horizontal placement. (**b**) Sideways placement.

**Table 1 sensors-22-04199-t001:** Structural parameters of the combination optical system.

Surface Type	Radius (mm)	Thickness (mm)	Glass	Diameter (mm)	Conic
Stop	0	5	Air	0	0
3	56.132	6	ZNSE	6.75	−112.603
4	Infinity	139.271	Air	6.75	0
6	Infinity	0	Air	6.75	0
7	123.491	6	ZNSE	6.75	−495.16
8	Infinity	317.448	Air	6.641	0

**Table 2 sensors-22-04199-t002:** Structural parameters of aspherical cylindrical-lens-combined optical system.

Surface Type	Radius (mm)	Thickness (mm)	Glass	Diameter (mm)	Conic
Stop	Infinity	5	Air	6.75	0
3	64.552	6	ZNSE	6.75	−64.672
4	Infinity	322.187	Air	6.75	0
6	123.491	6	ZNSE	6.75	−85.982
7	Infinity	10	Air	6.564	0
8	−124.613	5	ZNSE	5.803	0
9	Infinity	10	Air	5.778	0
11	266.913	6	ZNSE	5.665	0
12	Infinity	10	Air	5.636	0
14	Infinity	490	Air	5.523	0

**Table 3 sensors-22-04199-t003:** Structural parameters of spherical cylindrical-lens-combined optical focusing system.

Surface Type	4th Order Term	6th Order Term	8th Order Term	10th Order Term
3	5.586 × 10^−6^	−1.250 × 10^−8^	1.058 × 10^−12^	0
5	0	0	−90	0
6	5.712 × 10^−6^	−1.311 × 10^−8^	2.484 × 10^−11^	0
10	0	0	90	0
13	0	0	−90	0

**Table 4 sensors-22-04199-t004:** Geometric parameters of cylindrical lenses.

	Radius (mm)	Lens Orientation	Focal Length (mm)	Size (mm)	Conic	4th Order Coefficient
Aspherical Cylindrical Lens 1	64.55	X	46.00	25 × 25	−4.672	5.11633 × 10^−6^
Aspherical Cylindrical Lens 2	123.49	Y	88.00	60 × 25	−5.982	5.25724 × 10^−6^
Cylindrical Lens 3	−24.91	Y	−88.00	60 × 25	0	
Cylindrical Lens 4	266.91	X	190.20	60 × 25	0	

**Table 5 sensors-22-04199-t005:** Working distance and displacement parameters of the zoom lens system.

Working Distance (mm)	The Distance from the Second Aspherical Cylindrical Lens to the Third Cylindrical Lens (mm)	The Distance from the Second Aspherical Cylindrical Lens to the Fourth Cylindrical Lens (mm)
500	10.43	31.18
505	10.36	29.14
510	10.29	27.16
515	10.22	25.25
520	10.15	23.41
525	10.07	21.62
530	10.00	20.00
535	9.93	18.20

**Table 6 sensors-22-04199-t006:** Linear fitted displacement values of the zoom lens system.

Working Distance (mm)	The Distance from the Second Aspherical Cylindrical Lens to the Third Cylindrical Lens (mm)	The Distance from the Second Aspherical Cylindrical Lens to the Fourth Cylindrical Lens (mm)	The Fitted Distance from the Second Aspherical Cylindrical Lens to the Third Cylindrical Lens (mm)	The Fitted Distance from the Second Aspherical Cylindrical Lens to the Fourth Cylindrical Lens (mm)
500	10.43	31.18	10.43	31.17
505	10.36	29.14	10.36	29.14
510	10.29	27.16	10.29	27.17
515	10.22	25.25	10.22	25.26
520	10.15	23.41	10.15	23.42
525	10.07	21.62	10.07	21.63
530	10.00	20.00	10.00	19.91
535	9.93	18.20	9.93	18.25

**Table 7 sensors-22-04199-t007:** Sintering parameters of molded parts.

	Laser Device Output Power (W)	Working Power (W)	Spot Size (mm)	Scan Speed (mm/s)	Hatch Spacing (mm)	Layer Thickness (mm)
Flat-top line-shaped spot	93	15	0.1 × 0.5	2000	0.5	0.1
Gaussian spot	15	15	Φ0.1	2000	0.1	0.1

**Table 8 sensors-22-04199-t008:** Molding data of sample 1.

Placement Spot		Horizontal			Sideways		
Laser Type	Time	0°	30°	45°	0°	30°	45°
Flat-top line-shaped spot (s)	$\bar{T_{a1}}$	104	117	153	105	114	152
	$\bar{T_{b1}}$	454	467	507	1511	1515	1571
Gaussian spot (s)	$\bar{T_{a2}}$	503	512	503	501	532	579
	$\bar{T_{b2}}$	853	863	853	1914	1932	1984
Flat-top line-shaped spot/Gaussian spot (%)	$\bar{T_{a1}}/\bar{T_{a2}}$	20.7	22.8	30.4	21.0	21.4	26.3
	$\bar{T_{b1}}/\bar{T_{b2}}$	53.3	54.1	59.4	78.9	78.4	79.2

**Table 9 sensors-22-04199-t009:** Molding data of sample 2.

Sample Model and Placement		2a			2b		
Laser Type	Time	0°	30°	45°	0°	30°	45°
Flat-top line-shaped spot (s)	$\bar{T_{a1}}$	1673	1745	2521	1717	1768	2627
	$\bar{T_{b1}}$	3114	3181	2937	8927	8964	9710
Gaussian spot (s)	$\bar{T_{a2}}$	8113	8242	8265	8135	8404	8483
	$\bar{T_{b2}}$	9493	9705	9698	15,237	15,507	15,533
Flat-top line-shaped spot/Gaussian spot (%)	$\bar{T_{a1}}/\bar{T_{a2}}$	20.6	21.2	30.5	21.1	21.0	31.0
	$\bar{T_{b1}}/\bar{T_{b2}}$	32.8	32.8	30.3	58.6	57.8	62.5

## Data Availability

Not applicable.

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
