# Peer review of "Flat-Top Line-Shaped Beam Shaping and System Design"

_sensors, 2022, doi:10.3390/s22114199_

Round 1

Reviewer 1 Report

The results of modeling and experiment of an optical system for the formation of a laser beam with a quasi-uniform distribution of the field intensity are presented.
After reading the article, there are the following questions.
1. Why are expressions (1) and (2) equal to 1?
2. It is desirable to explain the choice of Galileo's scheme, its magnification, as well as the derivation of formula (9) (or a reference to the literature).
3. It is desirable to explain how the input Gaussian CO2 laser beam was set in Zemax.
4. It is desirable to explain why you use two cylindrical lenses with positions 3 and 4 in Figure 6, and not one cylindrical lens.

Author Response

1. Why are expressions (1) and (2) equal to 1?

The original expressions ( 1 ) and ( 2 ) are the results of the intensity normalization of the incident Gaussian beam and the outgoing flat-topped beam, which are now changed to Formulas ( 2 ) and ( 3 ).

2. It is desirable to explain the choice of Galileo's scheme, its magnification, as well as the derivation of formula (9) (or a reference to the literature).

References [20] have been added before expression (10), and the description of choosing Galileo scheme and the structure diagram of Galileo beam shaping system have been added after expression (10).

3. It is desirable to explain how the input Gaussian CO2 laser beam was set in Zemax.

In the beginning of 3. System Design of Laser Beam Expansion and Shaping, the parameters of Gaussian CO2 laser beam set in Zemax are : CO2 laser, power 200W, wavelength 10.6μm, incident spot diameter 9mm. The expression of Gaussian beam intensity distribution function is shown in the expression (4).  

4. It is desirable to explain why you use two cylindrical lenses with positions 3 and 4 in Figure 6, and not one cylindrical lens.

In Figure 6, cylindrical lens 3 and aspheric lens 1 are combined to shape and focus the beam in the y direction, while cylindrical lens 4 and aspheric lens 2 are combined to shape and focus the beam in the x direction. Therefore, two cylindrical lenses are needed, rather than a cylindrical lens. In the future research, for flat-top linear beam spots of other sizes, it is necessary to design laser beam expander shaping system and focusing system with different parameters. Therefore, the use of two cylindrical focusing mirrors is conducive to the design of laser beam shaping system in x direction and y direction with different parameters.

Reviewer 2 Report

Review for “Flat-top line-shaped beam shaping and system design”
The authors provide interesting insight into beam shaping techniques as well as relevant optimization procedures. Before recommending publication, the following observations have to be addressed:
1. Line 33: Current beam-shaping methods mainly include aspherical-lens systems [4-7], diffractive optical elements [8], and liquid-crystal spatial light modulators [9, 10]. Here, the authors should update their references with recent efforts in wave front control by means of metasurfaces and metamaterials ( https://www.sciencedirect.com/science/article/abs/pii/S0022407320305124?via%3Dihub) as well as others.
2. Figures 4 and 5 must be converted into tables and have their complete explanation rather than just 0.00 or 5.7e-6 etc…
3. There is no connection between the aspherical coefficients and the theoretical model of the flat-top beam. The authors should at least provide a section-view of the lenses in the classic cylindrical configuration and the aspherical configuration.
4. The light field distribution in Figure 3 has no reference image. Is this distribution proven to be good or bad and with respect to what?
5. The authors assume the CO2 laser provides a Gaussian beam at the entrance. This should be quantitatively proven at least by providing images of the beam as well as an M2 (msquared) measurements on Ox and Oy.
Based on these observations, I can only recommend for publication after a Major Revision is performed and all points are addressed.

Author Response

1. Line 33: Current beam-shaping methods mainly include aspherical-lens systems [4-7], diffractive optical elements [8], and liquid-crystal spatial light modulators [9, 10]. Here, the authors should update their references with recent efforts in wave front control by means of metasurfaces and metamaterials ( https://www.sciencedirect.com/science/article/abs/pii/S0022407320305124?via%3Dihub) as well as others.

I have added references [11,12].

2. Figures 4 and 5 must be converted into tables and have their complete explanation rather than just 0.00 or 5.7e-6 etc…

Figures 2, 4 and 5 have been converted into tables, as shown in Tables 1, 2 and 3.Table 3 is the aspheric coefficient structure parameters of lens 1 and lens 2.

3. There is no connection between the aspherical coefficients and the theoretical model of the flat-top beam. The authors should at least provide a section-view of the lenses in the classic cylindrical configuration and the aspherical configuration.

A description of ' The geometrical dimensions of aspherical cylindrical lens 1, aspherical cylindrical lens 2, cylindrical lens 3 and cylindrical lens 4 are shown in appendix. ' has been added to the last paragraph of 3.2.1

4. The light field distribution in Figure 3 has no reference image. Is this distribution proven to be good or bad and with respect to what?

Light distribution has been added as shown in Figure 4

5. The authors assume the CO2 laser provides a Gaussian beam at the entrance. This should be quantitatively proven at least by providing images of the beam as well as an M2 (msquared) measurements on Ox and Oy.

Gaussian laser intensity maps have been added at the beginning of 2. the Physical Model and Mapping Function of Flat-top Beam, such as expression (1), Figure (1)

Round 2

Reviewer 1 Report

After studying the manuscript with corrections, there are several questions.

1. In expression (1) r is the coordinate, r_0 is the beam radius in terms of the intensity level I_0/e^2. The same notation should be used for the beam radius r_0 or omega_0 in formulas (1), (4) and in figure (1).

2. In formula (4): 2/(pi*omega_0^2), exp[-2(r/omega_0)^2].

Author Response

1. In expression (1) r is the coordinate, r_0 is the beam radius in terms of the intensity level I_0/e^2. The same notation should be used for the beam radius r_0 or omega_0 in formulas (1), (4) and in figure (1).

The description of r is incorrect and has been changed.

The description of r_0 has been changed to “r_0 is the laser beam radius (mm).”.

The description of omega_0 is correct.

In Figure 1, r has been changed to r_0.

2. In formula (4): 2/(pi*omega_0^2), exp[-2(r/omega_0)^2].

Formula 4 has been modified.

Reviewer 2 Report

The authors have answered my questions in a satisfying manner, I recommend publication.

Author Response

Thank the reviewer for your affirmation of this article.